# Supra-Sartorial Subcutaneous Infiltration (SSSI) for Anterior Femoral Cutaneous Nerve Coverage in Total Knee Arthroplasty: A Preliminary Clinical Study

**DOI:** 10.3390/biomedicines13102368

**Published:** 2025-09-27

**Authors:** Shang-Ru Yeoh, Wei-Chun Chang, Kuan-Lin Wang, Kuang-Yu Tai, Fu-Kai Hsu, Ching-Wei Chuang

**Affiliations:** 1Department of Anesthesiology, Wan Fang Hospital, Taipei Medical University, Taipei 116, Taiwan; 2Center for Regional Anesthesia and Pain Medicine, Wan Fang Hospital, Taipei Medical University, Taipei 116, Taiwan; 3Department of Orthopedic Surgery, Wan Fang Hospital, Taipei Medical University, Taipei 116, Taiwan

**Keywords:** anterior cutaneous nerve of thigh, anterior femoral cutaneous nerve, intermediate femoral cutaneous nerve, multimodal analgesia, medial femoral cutaneous nerve, supra-sartorial subcutaneous infiltration, total knee arthroplasty

## Abstract

**Background**: Multimodal analgesia, combining adductor canal block (ACB) and local infiltration analgesia (LIA), is commonly used for pain control after total knee arthroplasty (TKA). However, ACB alone may not fully cover the anteromedial knee, a region extensively disrupted by TKA. Recent studies suggest that blocking branches of the anterior femoral cutaneous nerve (AFCN) could enhance analgesia, but targeted AFCN blocks are technically challenging. We evaluated supra-sartorial subcutaneous infiltration (SSSI) at the femoral triangle apex as a simpler alternative to AFCN blocks. **Methods**: We retrospectively reviewed 19 patients undergoing TKA with a standardized multimodal analgesic protocol, including intraoperative LIA limited to posterior capsule (PC-LIA), postoperative SSSI, and delayed intermittent ACB via catheter. SSSI involved infiltrating 20 mL of 0.3% ropivacaine into the subcutaneous plane above the sartorius muscle at the level of femoral triangle apex. Pain was assessed using Numerical Rating Scale (NRS) scores at rest and during movement at 9:00 PM on postoperative day 0 (POD 0) and 9:00 AM on POD 1, with scheduled ACB doses administered at the time of NRS pain score assessments. Rescue ACB boluses were given for intolerable pain before the first scheduled dose. **Results**: Eleven patients (58%) required no rescue analgesia before the first scheduled ACB, maintaining NRS scores ≤ 4 at rest and with movement for a minimum of 575–785 min post-spinal anesthesia. Eight patients needed rescue ACB, with variable pain relief. **Conclusions**: SSSI, when combined with PC-LIA, provided clinically meaningful analgesia in 58% of our patient cohort following TKA, though the variability observed suggests limited consistency. As a practical alternative to targeted AFCN blocks, SSSI could potentially complement ACB in multimodal pain management, but its efficacy remains uncertain due to the retrospective, non-controlled study design without a comparator group. Further investigation through prospective randomized controlled trials is warranted to validate these preliminary findings.

## 1. Introduction

A recent review concluded that the most effective analgesia after total knee arthroplasty (TKA) is achieved through a combination of the adductor canal block (ACB) and local infiltration analgesia (LIA), particularly when the posterior capsule is targeted (PC-LIA) [1]. This multimodal approach is intended to provide comprehensive coverage of the anteromedial and posterior aspects of the knee via ACB and PC-LIA, respectively.

In the anteromedial knee region that is surgically disrupted during TKA, cadaveric and volunteer studies by Bjørn et al. [2,3,4] suggest that the analgesic effect of ACB may be further enhanced by targeting two major branches of the anterior femoral cutaneous nerve (AFCN): the anterior branch of the medial femoral cutaneous nerve (MFCN-A) and the intermediate femoral cutaneous nerve (IFCN). However, the supporting clinical evidence to date is still very limited in the context of acute pain management after TKA. In addition, ultrasound identification of these cutaneous nerves is technically challenging because of their small caliber and variable course [5]. Precise confirmation often requires nerve tracking with a high-resolution probe, rendering targeted AFCN blocks impractical in routine perioperative practice.

Here, we present a retrospective case series evaluating the analgesic efficacy of supra-sartorial subcutaneous infiltration (SSSI) performed at the femoral triangle apex as a practical proxy to targeted AFCN blocks in the immediate postoperative period following TKA. This investigation serves as a preliminary step toward a planned randomized controlled trial comparing SSSI combined with intermittent ACB versus intermittent ACB alone.

## 2. Materials and Methods

At our institution, a multimodal analgesic protocol was collaboratively implemented for a subset of TKA patients, involving both anesthesiologists and orthopedic surgeons. Postoperative SSSI were performed by anesthesiologists, whereas PC-LIA was administered intraoperatively by surgeons. Intermittent ACBs delivered via a catheter were intentionally delayed and combined with non-opioid intravenous analgesics.

In the post-anesthesia care unit, SSSI was performed by infiltrating 20 mL of 0.3% ropivacaine from lateral to medial across the sartorius muscle at the femoral triangle apex, located sonographically at the intersection of the medial border of sartorius muscle and the medial border of adductor longus muscle. Using an inject-as-you-advance technique, local anesthetic was deposited evenly within the subcutaneous layer superficial to the fascia lata, while care was taken to avoid direct injection into hyperechoic structures resembling cutaneous nerves (Figure 1).

An adductor canal catheter was then inserted proximal-to-distally over a distance of 6–10 cm at the same level, without an initial local anesthetic bolus. The first scheduled ACB dose, consisting of 10 mL of 0.3% ropivacaine via the catheter, was administered at 9:00 PM on the day of surgery and repeated every 12 h, supplemented with intravenous parecoxib or propacetamol at the anesthesiologist’s discretion. PC-LIA consisted of two 10 mL injections of 0.3% ropivacaine delivered intraoperatively to the posteromedial and posterolateral aspects of the posterior capsule before cementation. The target injection plane was the potential space between the posterior capsule and the popliteal artery.

Numerical Rating Scale (NRS) pain scores at rest and during movement were recorded at each scheduled ACB administration time point. Rescue ACB boluses via the catheter were provided if patients experienced intolerable pain prior to the scheduled ACB dose, with corresponding NRS scores documented. All patients received spinal anesthesia with 11–13 mg of 0.5% bupivacaine, and only surgeries starting before noon were included to minimize confounding effects of residual spinal anesthesia.

Pain outcomes were summarized using medians and interquartile ranges (IQRs) due to the small sample size (*n* = 19) and potential non-normal distribution of the data. To assess the influence of demographic factors on pain outcomes, Spearman’s rank correlation was used to evaluate the relationship between age and NRS scores, and the Mann–Whitney U test was employed to compare NRS scores between sexes. The need for rescue ACB was analyzed as a binary outcome. Logistic regression was used to assess whether age or sex predicted the need for rescue ACB, and Fisher’s exact test was applied to examine the association between sex and rescue ACB requirement, given the small sample size and low frequency of male patients (*n* = 3). All statistical analyses were performed using R version 4.2.1, with a significance threshold of *p* < 0.05. Results were interpreted cautiously due to the limited sample size and retrospective study design.

## 3. Results

Among 19 TKA patients who received postoperative SSSI, 11 (58%) did not require rescue analgesia before the first scheduled ACB at 9:00 PM on the day of surgery. The interval from spinal anesthesia to the initial ACB ranged from 575 to 785 min, during which NRS pain scores remained ≤ 4 at rest and with movement. At 9:00 PM on POD 0, the median NRS scores in this group were 2.0 (IQR 1.0–2.5) at rest and 2.0 (IQR 1.0–3.0) with movement. The remaining 8 patients required rescue analgesia prior to the scheduled ACB, with variable reductions in NRS scores following rescue dosing. Their median NRS scores were 5.5 (IQR 5.0–7.0) at rest and 6.0 (IQR 5.0–8.0) with movement (Table 1).

Statistical analysis of POD 0 pain scores showed no significant effect of age or sex using Spearman correlation and Mann–Whitney U tests, indicating no influence of these demographic factors on initial pain outcomes. Logistic regression and Fisher’s exact test further revealed that age and sex did not predict the need for rescue ACB, suggesting minimal demographic impact in this cohort. These findings are limited by the small sam-ple size (*n* = 19, with only 3 males), requiring cautious interpretation.

It should also be emphasized that the design of this study does not allow determination of the full duration of action of SSSI. Because the ACB was administered at a predetermined time point, the observed effect reflects only the minimal duration of SSSI efficacy within the multimodal regimen. The actual duration of SSSI may extend beyond the measured interval, but this cannot be ascertained from the present data.

## 4. Discussion

By delaying the first scheduled ACB administration and reserving the catheter for rescue analgesia, our patient cohort demonstrated clinically meaningful, though variable, analgesic efficacy of the SSSI when combined with PC-LIA. Effective postoperative analgesia lasted a minimum of 575–785 min after spinal anesthesia and surgical initiation. Considering that sensory recovery from intrathecal bupivacaine typically occurs at approximately 212 ± 54 min [6], the prolonged pain relief observed cannot be explained by spinal anesthesia alone. And because PC-LIA likely shares similar local anesthetic coverage and effects with the infiltration between the popliteal artery and posterior capsule [7], a technique specifically targeting posterior knee pain, the observed analgesic effects to the anteromedial knee is best attributed to the SSSI. However, the absence of a control group precludes definitive attribution of the observed analgesic effects to SSSI alone.

Near the femoral triangle apex, the IFCN generally appears as one to three branches (most often two) [2,8], traveling within the subcutaneous layer [4,8,9] or enclosed in a fascia lata duplicature [2,3,4,10], superficial to the sartorius or vastus medialis muscles. In comparison, the MFCN-A main branch is often encased within a fascia lata duplicature [3,4,8] or located deep to the fascia lata [9] on the superficial surface of the sartorius muscle. Thus, subcutaneous infiltration superficial to the fascia lata (i.e., SSSI) may bypass the MFCN-A itself. However, an anatomical study has shown that branches of the IFCN and MFCN-A anastomose within the subcutaneous plane, forming a network of cutaneous rami on the anterior thigh [11]. Accordingly, the SSSI likely exerts its effects by blocking the IFCN main branches and the communicating rami of the MFCN-A—structures difficult to visualize even with high-resolution ultrasound, given their small caliber, fat-like echogenicity, and variable courses.

Although the IFCN is relatively consistent in its anatomical course, the MFCN exhibits substantial variability in its trajectory, branching pattern, and relationship to the sartorius muscle and fascia lata [8,9]. To effectively block these variable cutaneous branches simultaneously, several technically complex techniques have been proposed, including dual targeted fascia lata duplicature injections at the midpart and apex of the femoral triangle [2,3,4], three aliquot depositions across different anatomical spaces relative to the sartorius, fascia lata, and superficial femoral artery [9,10], and meticulous nerve tracking to their femoral nerve origins [8,12]. Consequently, targeted AFCN blocks—particularly the MFCN-A block—can be technically demanding and time-consuming, limiting their practicality in perioperative settings. In comparison, the SSSI used in our study requires only sonographic identification of the subcutaneous layer superficial to the sartorius muscle, followed by a single infiltration without needle redirection. Taken together with its demonstrated efficacy in our cohort, the SSSI appears better suited than targeted AFCN blocks for routine anesthesia practice.

Traditionally, ACB’s analgesic effect on the anteromedial knee has been attributed to blockade of the saphenous nerve (SN). Other relevant targets include the nerve to vastus medialis, which terminates as the medial retinacular nerve and may significantly influence postoperative pain [13]. Yet, findings by Bjørn et al. [2,3,4] and Kampitak et al. [9] indicate that ACB alone may provide incomplete coverage of the anteromedial knee in TKA. Importantly, the MFCN-A, IFCN, and SN contribute complementary sensory innervation to the midline incision of TKA, with marked inter-individual variability [3,12]. In fact, combined MFCN-A and IFCN contributions exceeded those of the SN in the majority of participants [3]. This sensory complementarity and substantial inter-individual variability may explain the inconsistent analgesic efficacy of SSSI observed in our cohort. Moreover, the MFCN-A innervates deeper anteromedial knee structures, including the medial retinaculum involved in the medial parapatellar arthrotomy [4]. Thus, by blocking the IFCN and at least part of the MFCN-A, the SSSI may complement ACB in covering both superficial cutaneous tissues and deeper genicular layers of the anteromedial knee affected by TKA.

This study’s findings are subject to several limitations that warrant cautious interpretation. The retrospective, non-randomized design and small sample size increase the risk of selection bias and limit the ability to establish causality or generalizability, particularly given the lack of a control group. Additionally, the variability in patient characteristics, such as sex distribution (only 3 males), and the inability to isolate SSSI’s independent duration of action due to the scheduled ACB timing further confound the results, emphasizing the need for larger, well-designed prospective trials to validate these preliminary observations.

## 5. Conclusions

Our findings suggest that SSSI, when combined with PC-LIA, may provide effective analgesia in approximately 60% of TKA patients during the immediate postoperative period. However, this effect is variable and should be interpreted with caution due to the retrospective, non-controlled study design and small sample size (*n* = 19). The observed duration of action of SSSI in our cohort represents only its minimal value, as the actual duration may be longer but cannot be determined within the current study design. Given that SSSI is performed near the femoral triangle apex—a site commonly utilized for ACB or distal femoral triangle block—combining these techniques may offer a rational approach to optimizing anteromedial knee analgesia. However, the practical benefits remain uncertain in the absence of a comparator group. As a potential alternative to targeted AFCN blocks, SSSI could serve as an easy-to-perform adjunct to ACB within multimodal pain management. Nevertheless, prospective randomized controlled trials are needed to confirm its efficacy and establish its role.

## Figures and Tables

**Figure 1 biomedicines-13-02368-f001:**
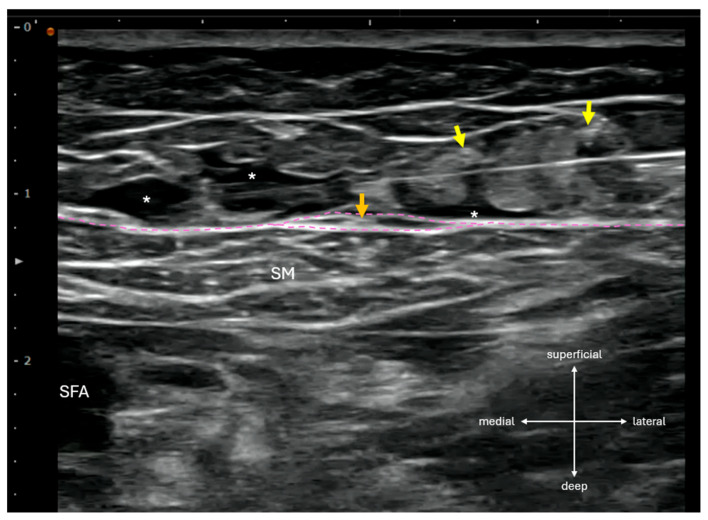
Supra-sartorial subcutaneous infiltration (SSSI) performed in a lateral-to-medial direction at the femoral triangle apex, corresponding to the level commonly used for adductor canal block (ACB) or distal femoral triangle block. On ultrasound, anterior femoral cutaneous nerve (AFCN) branches appear as small, hyperechoic, ovoid structures superficial to the sartorius muscle (SM). The intermediate femoral cutaneous nerve (IFCN, yellow arrows) typically presents as two main branches in the subcutaneous layer, while the anterior branch of the medial femoral cutaneous nerve (MFCN-A, orange arrow) may be encased within a fascia lata duplicature (magenta lines). The course and branching pattern of these nerves, particularly the MFCN-A, are highly variable and often difficult to identify sonographically. *: local anesthetic; SFA: superficial femoral artery.

**Table 1 biomedicines-13-02368-t001:** Outcomes from the patient cohort demonstrating the multimodal analgesic efficacy of supra-sartorial subcutaneous infiltration (SSSI) combined with posterior capsule local infiltration analgesia (PC-LIA), delayed adductor canal block (ACB), and non-opioid analgesics. Only Numerical Rating Scale (NRS) pain scores recorded at 9:00 PM on postoperative day 0 (POD 0) and 9:00 AM on POD 1 are shown, as data beyond POD 1 at 9:00 AM were not relevant to this study.

**No Request for Rescue ACB Before the Scheduled Dose**							
Patient Number	Age (y/o)	Sex	Time from SA to First Scheduled ACB (min)	NRS Pain Scores			
				POD 0 at 9:00 PM		POD 1 at 9:00 AM	
				Resting	Dynamic	Resting	Dynamic
1	76	F	785	1	1	2	2
2	73	F	785	1	1	2	2
3	47	F	780	2	2	2	2
4	75	M	780	1	1	0	1
5	77	M	780	1	1	0	2
6	75	F	680	3	4	1	2
7	57	F	655	2	2	1	2
8	71	F	650	2	2	0	1
9	88	F	645	2	3	2	2
10	72	F	645	3	3	3	3
11	76	F	575	3	4	4	4
**Requested for Rescue ACB Before the Scheduled Dose**							
Patient number	Age (y/o)	Sex	Time from SA to rescue ACB (min)	NRS pain scores			
				POD 0 at the time of rescue ACB		POD 1 at 9:00 AM	
				Resting	Dynamic	Resting	Dynamic
12	60	M	440	6	6	4	4
13	73	F	420	8	9	3	3
14	79	F	415	7	8	2	2
15	81	F	375	5	5	2	2
16	80	F	355	5	5	2	2
17	83	F	290	7	8	5	6
18	73	F	270	5	6	1	2
19	72	F	225	5	5	3	3

Abbreviations: F, female; M, male; min, minutes; NRS, numerical rating scale; y/o, years old.

## Data Availability

The raw data supporting the findings of this study are available from the corresponding author upon reasonable request.

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
