# Peer review of "Supra-Sartorial Subcutaneous Infiltration (SSSI) for Anterior Femoral Cutaneous Nerve Coverage in Total Knee Arthroplasty: A Preliminary Clinical Study"

_biomedicines, 2025, doi:10.3390/biomedicines13102368_

Round 1

Reviewer 1 Report

Comments and Suggestions for Authors

I read your communication with great interest. The topic is relevant for clinicians involved in perioperative pain management, as the anteromedial knee region often remains challenging to cover adequately after total knee arthroplasty. Your proposal of supra-sartorial subcutaneous infiltration (SSSI) as a simpler alternative to targeted anterior femoral cutaneous nerve blocks is both practical and original.

That said, the study has several limitations that should be addressed more openly in the discussion. First, the retrospective design and the small sample size (only 19 patients) restrict the strength of the conclusions. This point is briefly acknowledged, but I think it deserves stronger emphasis, especially when considering how generalizable the results may be. Second, because there was no control group, it is difficult to be certain how much of the observed effect can be attributed to SSSI itself as opposed to other elements of the multimodal regimen. It would be useful to explain this limitation more explicitly.

The results are presented clearly, and the inclusion of both resting and dynamic NRS scores is a strength. Still, I suggest that the authors interpret their findings with a bit more caution, avoiding overstatements. It may help to remind readers that the observed analgesia was variable across patients and that a significant proportion still required rescue ACB before the scheduled dose. Alongside this, I encourage you to expand the discussion by comparing your findings with other techniques described in recent literature, such as femoral triangle blocks, iPACK, or para-sartorial approaches. Doing so would place the present study in a broader clinical context.

Regarding methodology, I would also like to see clarification on whether any statistical analysis was attempted beyond descriptive reporting, and whether potential confounding factors (such as variations in surgical technique or comorbidities) might have influenced pain outcomes. Even if the dataset is too small for robust statistics, a comment on this point would strengthen transparency.

Overall, this is a promising preliminary study that may inspire future randomized controlled trials. With a more balanced discussion, a clearer statement of the study’s limitations, and minor refinements in language for fluency, the manuscript could be improved significantly and provide meaningful value to the readership

Author Response

[Comments 1] First, the retrospective design and the small sample size (only 19 patients) restrict the strength of the conclusions. This point is briefly acknowledged, but I think it deserves stronger emphasis, especially when considering how generalizable the results may be. Second, because there was no control group, it is difficult to be certain how much of the observed effect can be attributed to SSSI itself as opposed to other elements of the multimodal regimen. It would be useful to explain this limitation more explicitly.

[Response 1] We appreciate the reviewer’s feedback regarding the limitations of our study design. To address this, we have expanded the discussion section to emphasize the constraints imposed by the retrospective, non-controlled design and small sample size, particularly with respect to the generalizability of our findings. These considerations have also been incorporated into the rewritten conclusion sections in both the abstract and the main text. We have also clarified that the absence of a control group precludes definitive attribution of the observed analgesic effects to SSSI alone.

[Comment 2] The results are presented clearly, and the inclusion of both resting and dynamic NRS scores is a strength. Still, I suggest that the authors interpret their findings with a bit more caution, avoiding overstatements. It may help to remind readers that the observed analgesia was variable across patients and that a significant proportion still required rescue ACB before the scheduled dose. Alongside this, I encourage you to expand the discussion by comparing your findings with other techniques described in recent literature, such as femoral triangle blocks, iPACK, or para-sartorial approaches. Doing so would place the present study in a broader clinical context.

[Response 2] We thank the reviewer for acknowledging the clarity of our results and the inclusion of both resting and dynamic NRS scores. To address concerns about potential overstatements, as in Response 1, we have revised the conclusion sections in both the abstract and main text to underscore the variability in analgesic outcomes. Regarding comparisons with other regional techniques (e.g., femoral triangle blocks, iPACK, or para-sartorial approaches), we agree that such comparisons would provide valuable context. However, given the retrospective design and limited sample size, our data may not support robust comparisons with these techniques. We have noted this limitation in the discussion and propose that such comparisons be explored in a prospective randomized controlled trial, which our team is currently designing.

[Comment 3] Regarding methodology, I would also like to see clarification on whether any statistical analysis was attempted beyond descriptive reporting, and whether potential confounding factors (such as variations in surgical technique or comorbidities) might have influenced pain outcomes. Even if the dataset is too small for robust statistics, a comment on this point would strengthen transparency.

[Response 3] We appreciate the reviewer’s suggestion to enhance transparency regarding statistical analyses and potential confounders. To address this, we have added non-parametric measures, namely, the medians and interquartile ranges (IQRs) of NRS pain scores on postoperative day 0 for both the no-rescue and rescue groups, to the results section. Additionally, we have included a new paragraph in the results section describing exploratory statistical analyses examining the influence of age and sex on pain outcomes.

Additional revisions made to the manuscript are as follows:

  • Line 20: In the abstract, the phrase “at the level of femoral triangle apex” was added to more precisely specify the anatomical location of SSSI.
  • Line 22: In the abstract, the phrase “with scheduled ACB doses administered at the time of NRS pain score assessments” was added to clarify that NRS pain scores were recorded concurrently with scheduled ACB administration.
  • Lines 100-110: A paragraph describing the methods of statistical analysis has been added.
  • Line 158: "near the femoral triangle apex" was deleted for its redundency
  • Lines 167-172: A new sentence was added to concisely describe all literature-reported anterior femoral cutaneous nerve (AFCN) blocking techniques, highlighting the technical simplicity of SSSI in contrast. The original sentence referencing the nerve tracking technique was removed, with its content integrated into the new sentence.
  • Line 184: An explicit citation of Kampitak et al. was added alongside Bjorn et al. to acknowledge the authors of the only randomized controlled trial investigating the effect of AFCN blocks in addition to a distal femoral triangle block in TKA patients.
  • Line 193: The word “may” was introduced to adopt a more cautious tone when discussing the potential analgesic effects of SSSI as a complement to ACB.
  • The legends for Table 1 were shortened to enhance readability.

Reviewer 2 Report

Comments and Suggestions for Authors

This work is written well, but since the sample size is not large, and therefore, I would suggest making a more accurate statement about the Supra-Sartorial Subcutaneous Infiltration (SSSI) in the Conclusion. For example, in the Abstract and Conclusion, you mention 60%, whereas in the manuscript the number is different, namely 11/19 = 57.9%.

In Conclusion, there is a statement: "Our findings suggest that the SSSI, when combined with PC-LIA, provides effective analgesia in approximately 60% of TKA patients during the immediate postoperative period." I would recommend to change 60% to 58% and also to reformulate the sentence; similar to a sentence in the Abstract: "SSSI, when combined with PC-LIA, provided clinically meaningful but variable analgesia in approximately [58%] of patients after TKA." As you understand, 'provided' is different from 'provides', because in the Abstract you show the results based on a small sample of patients n = 19, whereas in the Conclusion the statement is more affirmative, although it is not guaranteed that another sample will have the similar percentage (i.e. 58%).

Also, I would suggest showing the range of NRS, e.g. NRS (0-10 scale).

Author Response

[Comment 1] This work is written well, but since the sample size is not large, and therefore, I would suggest making a more accurate statement about the Supra-Sartorial Subcutaneous Infiltration (SSSI) in the Conclusion. For example, in the Abstract and Conclusion, you mention 60%, whereas in the manuscript the number is different, namely 11/19 = 57.9%.

[Response 1] We appreciate the reviewer’s feedback regarding the limitations of our study design. To address this, we have expanded the discussion section to emphasize the constraints imposed by the retrospective, non-controlled design and small sample size, particularly with respect to the generalizability of our findings. These considerations have also been incorporated into the rewritten conclusion sections in both the abstract and the main text. We have also clarified that the absence of a control group precludes definitive attribution of the observed analgesic effects to SSSI alone.

[Comment 2] In Conclusion, there is a statement: "Our findings suggest that the SSSI, when combined with PC-LIA, provides effective analgesia in approximately 60% of TKA patients during the immediate postoperative period." I would recommend to change 60% to 58% and also to reformulate the sentence; similar to a sentence in the Abstract: "SSSI, when combined with PC-LIA, provided clinically meaningful but variable analgesia in approximately [58%] of patients after TKA." As you understand, 'provided' is different from 'provides', because in the Abstract you show the results based on a small sample of patients n = 19, whereas in the Conclusion the statement is more affirmative, although it is not guaranteed that another sample will have the similar percentage (i.e. 58%).

[Response 2] We have revised the results sections in both the abstract and main text to consistently report the proportion of patients experiencing effective analgesia as 58%, reflecting the precise calculation (11/19). In the main text conclusion, we rephrased the statement to: “Our findings suggest that SSSI, when combined with PC-LIA, may provide effective analgesia in approximately 60% of TKA patients during the immediate postoperative period.” The term “may” was introduced to adopt a more cautious tone regarding the potential analgesic effects of SSSI as a complement to ACB, while “approximately 60%” was retained to acknowledge the limited generalizability of our findings due to the small sample size and retrospective design.

[Comment 3] Also, I would suggest showing the range of NRS, e.g. NRS (0-10 scale).

[Response 3] We agree with the reviewer that clarifying the NRS range enhances transparency. Given the small sample size, we opted for non-parametric measures, as they are more robust for potentially skewed data. Accordingly, we have added medians and interquartile ranges (IQRs) of NRS pain scores on postoperative day 0 for both the no-rescue and rescue groups to the results section. Additionally, a new paragraph in the results section describes exploratory statistical analyses examining the influence of age and sex on pain outcomes, further addressing potential variability in the data.

Additional revisions made to the manuscript are as follows:

  • Line 20: In the abstract, the phrase “at the level of femoral triangle apex” was added to more precisely specify the anatomical location of SSSI.
  • Line 22: In the abstract, the phrase “with scheduled ACB doses administered at the time of NRS pain score assessments” was added to clarify that NRS pain scores were recorded concurrently with scheduled ACB administration.
  • Lines 100-110: A paragraph describing the methods of statistical analysis has been added.
  • Line 158: "near the femoral triangle apex" was deleted for its redundency
  • Lines 167-172: A new sentence was added to concisely describe all literature-reported anterior femoral cutaneous nerve (AFCN) blocking techniques, highlighting the technical simplicity of SSSI in contrast. The original sentence referencing the nerve tracking technique was removed, with its content integrated into the new sentence.
  • Line 184: An explicit citation of Kampitak et al. was added alongside Bjorn et al. to acknowledge the authors of the only randomized controlled trial investigating the effect of AFCN blocks in addition to a distal femoral triangle block in TKA patients.
  • Line 193: The word “may” was introduced to adopt a more cautious tone when discussing the potential analgesic effects of SSSI as a complement to ACB.

The legends for Table 1 were shortened to enhance readability.

Reviewer 3 Report

Comments and Suggestions for Authors

Dear Authors,

Thank you for the opportunity to review your manuscript. This preliminary clinical study addresses a very relevant and practical issue in postoperative pain management for total knee arthroplasty. Your work on Supra-Sartorial Subcutaneous Infiltration (SSSI) as a simplified alternative to targeted anterior femoral cutaneous nerve blocks is a valuable contribution that provides a foundation for future research.

I have summarized the key points of your study, including its strengths and areas for potential improvement, to help you enhance the manuscript.

 The study tackles a common clinical problem—incomplete anteromedial knee analgesia after TKA with standard adductor canal blocks (ACB) and local infiltration analgesia (LIA). The SSSI technique you propose is described as a practical and easy-to-perform alternative to the more technically demanding targeted AFCN blocks.

The anatomical and physiological basis for the SSSI approach is well-articulated, explaining how it complements ACB by targeting nerves that provide sensory innervation to the anteromedial knee. This provides a strong theoretical foundation for your intervention.

Clear and Concise Presentation: The manuscript is well-structured and easy to follow. The results are presented clearly, and the discussion effectively links the findings to existing literature.

While your study provides compelling preliminary data, its retrospective and non-comparative design is a significant limitation that must be more explicitly discussed. To strengthen your manuscript, I recommend you address the following points:

 The study includes a small sample of only 19 patients. While this is acceptable for a preliminary case series, it limits the generalizability of your findings. I suggest you clearly state this limitation in your discussion, as the observed analgesic efficacy of approximately 60% might not be representative of a larger patient population.

 The absence of a control group (e.g., a group receiving only PC-LIA and ACB without SSSI) makes it difficult to definitively attribute the analgesic effects to the SSSI procedure alone. Although you note that prolonged pain relief cannot be explained by spinal anesthesia alone, a direct comparison would be needed to establish the independent effect of SSSI. You could consider using a more cautious tone in the abstract and conclusion to reflect this.

You appropriately highlight that the analgesic effect of SSSI was "clinically meaningful but variable". This variability warrants further exploration. A more detailed analysis of the characteristics of the 8 patients who required rescue analgesia could provide valuable insights. For example, were there differences in their age, gender, type of surgery, or other comorbidities that might explain why SSSI was less effective for them? Adding this analysis would enrich the manuscript.

You correctly point out that the study design does not allow you to determine the full duration of the SSSI's analgesic effect. This is a crucial limitation that should be emphasized more prominently in your conclusion, as it impacts the interpretation of the "prolonged" pain relief observed.

The legend for Table 1 is quite long and could be more concise. Consider reducing the text to make it more succinct and improve the overall readability of the table.

In summary, the manuscript presents promising preliminary findings for a practical analgesic technique. However, its retrospective nature and small sample size are significant limitations. I recommend that you incorporate a more thorough discussion of these points, especially the potential sources of variability in patient outcomes, to provide a more balanced and robust account of your findings. The authors' statement that this study serves as a preliminary step for a planned randomized controlled trial is an important aspect of this manuscript.

Author Response

[Comment 1] The study includes a small sample of only 19 patients. While this is acceptable for a preliminary case series, it limits the generalizability of your findings. I suggest you clearly state this limitation in your discussion, as the observed analgesic efficacy of approximately 60% might not be representative of a larger patient population.

 The absence of a control group (e.g., a group receiving only PC-LIA and ACB without SSSI) makes it difficult to definitively attribute the analgesic effects to the SSSI procedure alone. Although you note that prolonged pain relief cannot be explained by spinal anesthesia alone, a direct comparison would be needed to establish the independent effect of SSSI. You could consider using a more cautious tone in the abstract and conclusion to reflect this.

[Response 1] We thank the reviewer for the constructive feedback on the study’s limitations. To address these concerns, we have expanded the discussion section to explicitly highlight the limitations of the retrospective, non-controlled design and small sample size, emphasizing their impact on the generalizability of our findings. We have also clarified in the discussion that the absence of a control group prevents definitive attribution of the observed analgesic effects to SSSI alone, distinct from other components of the multimodal regimen. To reflect a more cautious tone, we have revised the conclusions in both the abstract and main text to underscore the preliminary nature of the findings and the need for further validation through controlled studies.

[Comment 2] You appropriately highlight that the analgesic effect of SSSI was "clinically meaningful but variable". This variability warrants further exploration. A more detailed analysis of the characteristics of the 8 patients who required rescue analgesia could provide valuable insights. For example, were there differences in their age, gender, type of surgery, or other comorbidities that might explain why SSSI was less effective for them? Adding this analysis would enrich the manuscript.

[Response 2] We appreciate the reviewer’s suggestion to explore the variability in analgesic outcomes. To address this, we have added a new paragraph in the results section presenting exploratory statistical analyses of the influence of age and sex on pain outcomes for the no-rescue and rescue groups. While no significant differences were identified, we have noted in the discussion that these findings are limited by the small sample size and should be interpreted cautiously. Due to the retrospective design and limited data, other potential confounders such as surgical technique or comorbidities could not be fully analyzed, but we propose further exploration in future prospective randomized controlled trial, which our team is currently designing

[Comment 3] You correctly point out that the study design does not allow you to determine the full duration of the SSSI's analgesic effect. This is a crucial limitation that should be emphasized more prominently in your conclusion, as it impacts the interpretation of the "prolonged" pain relief observed.

[Response 3] We agree with the reviewer on the importance of emphasizing the limitation regarding the duration of SSSI’s analgesic effect. To address this, we have revised the conclusion in the main text to state: “The observed duration of action of SSSI in our cohort represents only its minimal value, as the actual duration may be longer but cannot be determined within the current study design.”

[Comment 4] The legend for Table 1 is quite long and could be more concise. Consider reducing the text to make it more succinct and improve the overall readability of the table.

[Response 4]  We have shortened the legend for Table 1 to improve conciseness and readability, as suggested.

Additional revisions made to the manuscript are as follows:

  • Line 20: In the abstract, the phrase “at the level of femoral triangle apex” was added to more precisely specify the anatomical location of SSSI.
  • Line 22: In the abstract, the phrase “with scheduled ACB doses administered at the time of NRS pain score assessments” was added to clarify that NRS pain scores were recorded concurrently with scheduled ACB administration.
  • Lines 100-110: A paragraph describing the methods of statistical analysis has been added.
  • Line 158: "near the femoral triangle apex" was deleted for its redundency
  • Lines 167-172: A new sentence was added to concisely describe all literature-reported anterior femoral cutaneous nerve (AFCN) blocking techniques, highlighting the technical simplicity of SSSI in contrast. The original sentence referencing the nerve tracking technique was removed, with its content integrated into the new sentence.
  • Line 184: An explicit citation of Kampitak et al. was added alongside Bjorn et al. to acknowledge the authors of the only randomized controlled trial investigating the effect of AFCN blocks in addition to a distal femoral triangle block in TKA patients.
  • Line 193: The word “may” was introduced to adopt a more cautious tone when discussing the potential analgesic effects of SSSI as a complement to ACB.
  • The legends for Table 1 were shortened to enhance readability.

Round 2

Reviewer 3 Report

Comments and Suggestions for Authors

Dear Authors,

I have reviewed the revised version of the manuscript “Supra-Sartorial Subcutaneous Infiltration (SSSI) for Anterior Femoral Cutaneous Nerve Coverage in Total Knee Arthroplasty: A Preliminary Clinical Study.”

I would like to commend you for your  thorough and constructive revisions in response to my previous comments.

  • The discussion and conclusion now clearly acknowledge the limitations of the retrospective design, small sample size, and absence of a control group, adopting a more appropriately cautious tone.

  • An exploratory analysis of demographic factors (age, sex) was added to investigate variability in analgesic response, with correct emphasis on the interpretative limits given the small cohort.

  • The conclusion was revised to better highlight the uncertainty regarding the true duration of SSSI’s analgesic effect.

  • Table legends were shortened, and the text was streamlined to improve clarity.

  • Additional improvements include refinement of the anatomical description, inclusion of Kampitak et al. alongside prior references, and a more precise methods section.

Taken together, these revisions substantially improve the manuscript. You have demonstrated responsiveness to peer review and have appropriately framed your findings as preliminary, requiring confirmation in prospective controlled trials.

In my opinion, the manuscript is now suitable for publication in Biomedicines in its present form.